# Spatial Patterns of Anthrax Outbreaks and Cases among Livestock in Lesotho, 2005–2016

**DOI:** 10.3390/ijerph17207584

**Published:** 2020-10-19

**Authors:** Relebohile Juliet Lepheana, James Wabwire Oguttu, Daniel Nenene Qekwana

**Affiliations:** 1Section Veterinary Public Health, Department of Paraclinical Science, Faculty of Veterinary Sciences, University of Pretoria, Pretoria 0110, South Africa; relebohilelepheana@gmail.com; 2Department of Livestock Services, Ministry of Agriculture and Food Security, Maseru 100, Lesotho; 3Department of Agriculture and Animal Health, College of Agriculture and Environmental Sciences, University of South Africa, Florida Science Campus, Johannesburg 1709, South Africa; joguttu@unisa.ac.za

**Keywords:** spatial, anthrax, livestock, districts, villages, Lesotho

## Abstract

Background: Although anthrax occurs globally, the burden of the disease remains particularly high in Africa. Furthermore, the disease anthrax has significant public health and economic implications. However, sufficient attention has not been given to the geographic distribution of anthrax outbreaks and cases in Lesotho. Therefore, this study investigates the spatial patterns of anthrax outbreaks and cases among livestock in Lesotho from 2005 to 2016. Methods: A cross-sectional study design was adopted to realise the objectives of this study using retrospective data of anthrax outbreaks and cases recorded by the Department of Livestock Services (DLS) between 2005 and 2016. Anthrax outbreaks were geo-coded at village level and aggregated at district level. Proportions and 95% CI of anthrax outbreaks and cases by village and district were calculated. Cartographic maps displaying the distribution of anthrax outbreaks and cases at village and district level were constructed. Results: A total of 38 outbreaks were reported over the study period, and they were clustered in the Lowlands districts of Lesotho. Most outbreaks (52.6%, 20/38) in livestock were reported in the Maseru district. The Leribe district reported the lowest proportions of outbreaks (5.3%, 2/38) and cases (0.6%, 3/526). At the village level, 18% (7/38) of outbreaks were in Maseru Urban, followed by Ratau (16%, 6/38) and Mofoka (13%, 5/38). The Maseru district reported the highest (1.3%, 369/29,070) proportion of cases followed by Mafeteng (0.9%, 73/8530). The village with the most cases was Kolo (10.5%, 21/200), followed by Thaba-Chitja (7.7%, 33/430). Conclusion: Anthrax outbreaks and cases exclusively occur in the Lowlands districts of Lesotho, with villages such as Mahobong, Pitseng, Kolo, and Thaba-Chitja having a higher risk of anthrax disease. Findings of the present study have serious public health implications in light of the fact that between 2003 and 2008 Lesotho’s main abattoir was closed; hence, most of the meat in Lesotho was imported and/or sourced from the informal slaughter facilities. Much larger studies are needed to further investigate factors contributing to spatial disparities in anthrax outbreaks and cases observed in this study. Findings of the present study can be used to guide the formulation of a policy on prevention and control of anthrax in Lesotho.

## 1. Introduction

Anthrax disease occurs globally with the burden being higher in Africa [1]. It is caused by *Bacillus anthracis*, a Gram-positive, aerobic, endospore-forming bacilli [2]. Anthrax spores can persist in the environment for a long period due to their ability to resist desiccation, gamma radiation, and disinfection [3]. This ability is responsible for perpetuating outbreaks in the endemic foci [2,4]. 

All warm-blooded animals are at risk of anthrax; however, herbivores [5,6], particularly cattle and sheep, are at a higher risk of contracting the disease [6,7,8,9]. Anthrax has also been reported in wildlife [10,11]. Clinical signs of anthrax reported in animals include swelling of the neck region, bleeding from all-natural orifices, bloating, and a rapidly putrefying carcass without rigor mortis setting in [2]. Human infections occur through either direct or indirect contact with infected animal carcasses or their products [12].

Areas with neutral or slightly alkaline soil, elevated temperatures, and humidity have been reported to have a high incidence of anthrax outbreaks [7]. In the USA, clusters of anthrax cases have been reported in the east, west, and south of Georgia [13]. In addition, several studies indicate that the disease mainly occurs in low-lying depressions [4,6,14].

In Lesotho, anthrax disease has implications for both public health and the trade in animals and animal products [9,15]. The public health importance of anthrax is exacerbated by the fact that from 2003 to 2008 Lesotho did not have an operational formal abattoir, with most meat sourced from informal slaughterhouses [15]. Furthermore, according to Seeiso and McCrindle [15], between 2004 and 2008 human anthrax deaths associated with the consumption of informally slaughtered livestock meat were reported in Lesotho. 

Lepheana et al. [9] hypothesised that anthrax outbreaks in Lesotho have spatial distribution. However, there is no published studies on geographical disparities in anthrax outbreaks in Lesotho. Therefore, the present study investigates the spatial distribution of anthrax outbreaks and cases among livestock in Lesotho and identifies populations at risk of infection. Results of this study provide a framework for the formulation of a spatial risk-based approach for the control of anthrax and the development of policies that can mitigate the impact of the disease on the health of the public and the trade of animal and animal products in Lesotho. 

## 2. Materials and Methods 

### 2.1. Study Area

The study was conducted in Lesotho, which is in the southern part of Africa and landlocked in South Africa. It lies between latitudes 28° and 31° South and longitudes 27° and 31° East. The entire country is about 30,355 km^2^ in area and lies over 1000 m above sea level. Lesotho is divided into four geographical zones: the Lowlands (1400–1800 m), the Foothills (1800–2000 m), Senqu Valley (1400–1800 m), and the Highlands (2000–3400 m). For purposes of this study, the country was divided into two main topographical zones: the Lowlands (1400 to 2000 m) and the Highlands (2000 to 3400 m) (Figure 1). Lesotho has 10 districts, made up of 260 villages (Figure 2 and Appendix A).

### 2.2. Data Source 

The study used retrospective data of all anthrax outbreaks and cases recorded by the Department of Livestock Services (DLS) from 2005 to 2016. In Lesotho, it is mandatory for field officers and farmers to report all suspected anthrax cases (Government of Lesotho 1969). Confirmation of suspected anthrax cases is based on clinical presentation and microscopic examination of blood smears by Maseru Central Veterinary Laboratory at DLS. An anthrax case is suspected whenever animals die unexpectedly showing one or more of the following signs: bloody discharge from natural orifices (ear, nose, mouth, anus), bloating and swelling of the lower abdomen (oedema). 

Confirmed cases of anthrax are recorded by the Epidemiology Unit of DLS using the Transboundary Animal Disease Information System (TADIS. The same information is uploaded onto the World Animal Health Information System of the World Organisation for Animal Health (OIE-WAHIS) and the Animal Resources Information System (ARIS) of the African Union Inter African Bureau for Animal Resources (AU-IBAR), as part of the formal animal disease reporting process. An outbreak, as defined by DLS, is an occurrence of one or more cases of anthrax in a location.

### 2.3. Data Management and Analysis 

The following variables were retrieved from the database: year, region, district, village, outbreak, case, and population at risk. The population at risk was defined as the total number of animals in a herd at the time of the outbreak. Anthrax outbreaks and cases were geo-coded at village level and aggregated at the district level. The data set was assessed for inconsistencies, including incorrect coordinates. Where incorrect coordinates were noted, data sets were joined using the closest polygon to the point of interest. This was only noted once in the case of Likotsi village of the Maseru district, which was joined to Maseru Urban.

Proportions of anthrax outbreaks and cases by village and district, and their corresponding 95% confidence intervals, were calculated using SAS 9.4 (SAS Institute Inc., Cary, NC, USA). Cartographic maps were used to display distributions of anthrax outbreaks and cases at village and district spatial scale using ArcGIS 10.5 (Esri 2016, Redlands, CA, USA). Due to the low sample size, further spatial statistical analyses could not be performed.

## 3. Results

### 3.1. Anthrax Outbreaks

All 38 anthrax outbreaks were reported only in the Lowlands districts, with no single outbreak reported in the Highlands. Within the Lowlands districts, the highest proportion of outbreaks was recorded in the Maseru district (53%). The Leribe district reported the lowest proportion of outbreaks (6%) (Table 1, Figure 3). At the village level, 18% of outbreaks were reported in Maseru Urban, 16% in Ratau, and 13% in Mofoka (Table 1, Figure 4).

### 3.2. Anthrax Cases

Five hundred and twenty-six cases of anthrax were reported in Lesotho from 2005 to 2016. Moreover, all these cases were reported only in the Lowlands districts of the country (Figure 4 and Figure 5).

The Leribe district had the highest (14.3%) proportion of cases, while the Mohale’s Hoek district reported the lowest proportion of cases (0.2%) (Table 2). Among the villages, Mahobong had the highest proportion of cases (20.0%), followed by Pitseng (12.5%) and Kolo (10.5%) (Table 2 and Figure 6).

## 4. Discussion

Spatial analysis has been used in epidemiological studies to investigate the clustering of disease outbreaks and identify areas or populations at a higher risk of infection [16,17]. In this study, we investigated the spatial distribution of anthrax outbreaks and cases in Lesotho. 

Only the districts and villages located in the Lowlands experienced outbreaks and cases. There were no outbreaks or cases reported in the districts or villages located in the Highlands. This pattern where only the Lowlands experienced outbreaks and cases could be attributed to the fact that the Lowlands areas of Lesotho are warmer than the Highlands [18,19]. Warm conditions are known to favour the survival of anthrax spores. This is confirmed by studies that have reported that warmer areas tended to experience more anthrax outbreaks compared to colder areas [4].

Dragon et al. [4] in Canada, Chen et al. [7] in China, and Chikerema et al. [20] in Zimbabwe observed that the highest proportions of anthrax outbreaks tended to occur in the low-lying areas of their respective countries. Outbreaks in the low-lying areas are attributed to the fact that they are more susceptible to flooding [21]. This is because during flooding anthrax spores get unearthed leading to an increased risk of anthrax outbreaks [2,4,11]. Therefore, it is possible that, even in Lesotho, the pattern of anthrax outbreaks observed in the Lowlands was similar to that observed in Canada, Zimbabwe, and China. However, due to lack of data, this could not be confirmed. In view of this, further studies are needed to investigate the role of climatic conditions in the distribution of the disease within the low-lying areas.

In view of the fact that outbreaks and cases exclusively occurred in the Lowlands, the authors are of the view that the introduction of a zoning system for anthrax control in Lesotho could mitigate the economic impact of anthrax outbreaks, associated with the ban of the sale of wool and mohair products on the international market during anthrax outbreaks [22]. Implementing a zoning system would involve permitting the trade in livestock and livestock products during outbreaks from only the Highlands, where cases have not been reported and the risk is very minimal or nonexistent, while implementing a ban of sale of animal and animal products from the Lowlands where the risk is high [23].

Anthrax outbreaks and cases in this study differed by districts and villages. The Maseru district and its associated villages recorded the highest proportions of outbreaks and cases. These could be attributed to the lapse in anthrax control strategies, differences in animal health management programmes, and socio-economic dynamics [9]. For example, the Department of Livestock Marketing holds livestock auctions in the 10 districts of Lesotho once every month [24]. However, sheep and cattle dealers usually avoid these price-regulated points and instead opt to bring livestock to Maseru [25], which is the most affluent of all the 10 districts and, in addition, it is the capital city of Lesotho. However, it is not possible to confirm how this contributes to the high number of cases in Maseru due to the lack of a system to trace the origin of animals and consequently the cases. 

Furthermore, the fact that the Maseru district and its associated villages recorded the highest proportions of outbreaks and cases suggests that there is a heightened risk of exposure to anthrax for people in these areas. This is appreciated when consideration is given to the fact that between 2003 and 2008, Lesotho’s only abattoir was not functional, and more than 80% of the meat sold in butchers came mainly from informal slaughter [15]. In addition, it is not uncommon for animals that have died due to known or unknown causes to be consumed by humans [9]. In view of this, the Ministry of Health and the Ministry of Agriculture and Food Security should jointly develop and implement anthrax awareness campaigns to strengthen public awareness of the disease in Lesotho.

This study is subject to the following limitations: First, the authors had no control over the quality of data collected, which is an inherent challenge of retrospective studies. In addition, due to low numbers at village level, spatial analysis such as global and local indicators of spatial autocorrelation to assess for clustering of cases could not be performed. Nevertheless, this study provides baseline information on the spatial patterns of anthrax outbreaks in Lesotho. This information can be used to guide policy formulation regarding anthrax control strategies.

## 5. Conclusions

Anthrax outbreaks and cases are restricted to the Lowlands and warmer districts of Lesotho, with the risk of the disease being higher in villages in the Maseru district. Therefore, the residents of these areas are at a higher risk of contracting anthrax. In view of this, vaccination and other control strategies should focus on the Lowlands areas where the disease outbreaks tend to exclusively occur. Since the outbreaks are exclusively limited to the Lowlands, the Lesotho DLS should consider introducing zoning as a control strategy for anthrax. Given the limited nature of this study, further research is needed to identify drivers for anthrax outbreaks in Maseru specifically and Lesotho. The findings of the present study can be used to guide the formulation of a policy on prevention and control of anthrax in Lesotho.

## Figures and Tables

**Figure 1 ijerph-17-07584-f001:**
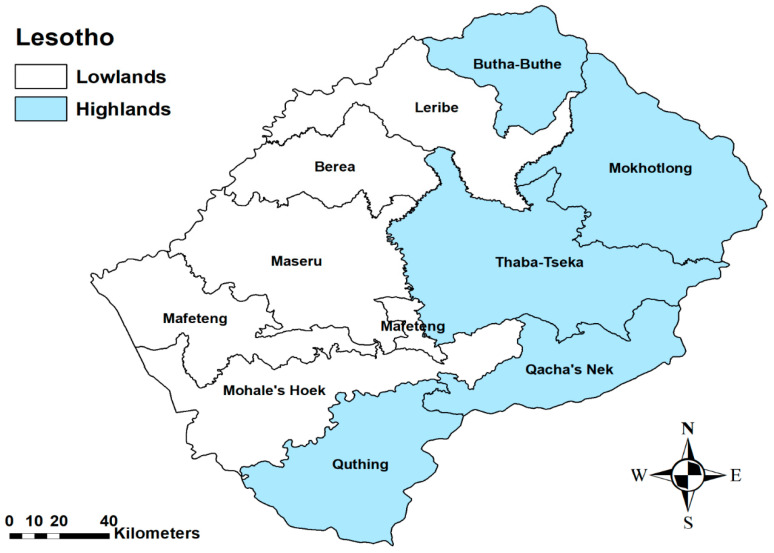
Map showing the ten administrative districts of Lesotho and the two topographical zones [10].

**Figure 2 ijerph-17-07584-f002:**
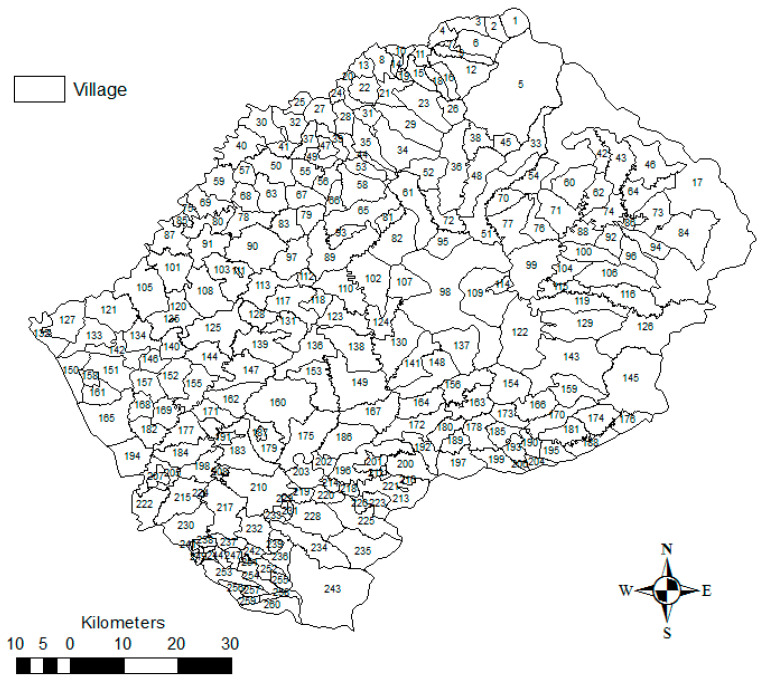
Map of Lesotho showing the boundaries of the smallest administrative areas, villages (see Appendix A for corresponding village names).

**Figure 3 ijerph-17-07584-f003:**
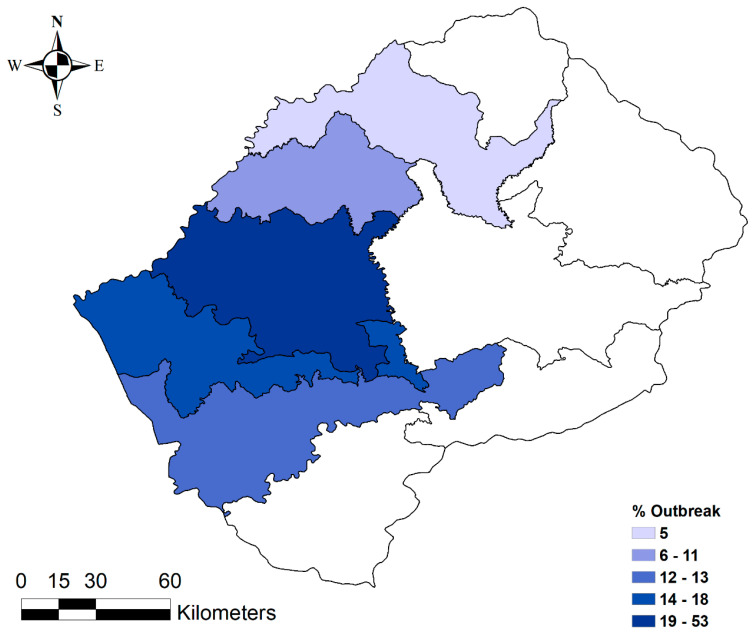
Distribution of anthrax outbreaks in the ten (*n* = 10) districts of Lesotho from 2005 to 2016.

**Figure 4 ijerph-17-07584-f004:**
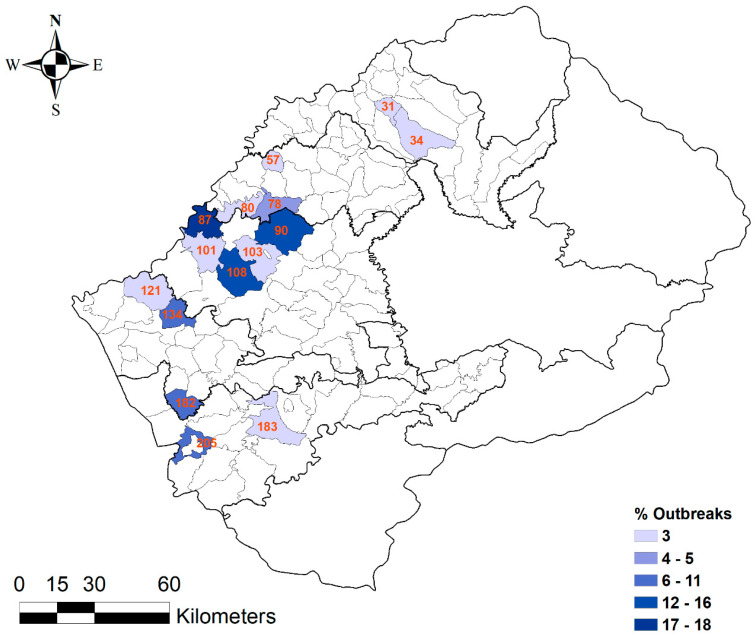
Distribution of anthrax outbreaks among the villages in the Lowlands of Lesotho from 2005 to 2016.

**Figure 5 ijerph-17-07584-f005:**
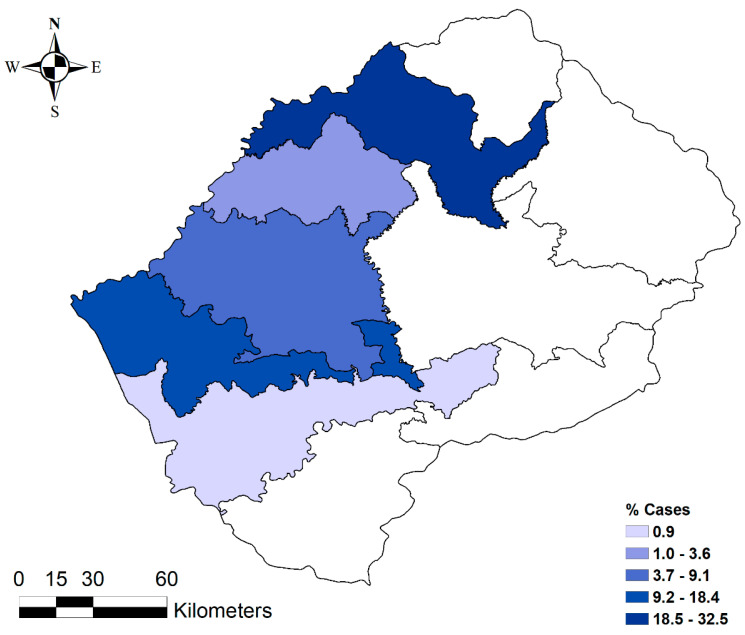
Distribution of anthrax cases in five Lowlands districts of Lesotho from 2005 to 2016.

**Figure 6 ijerph-17-07584-f006:**
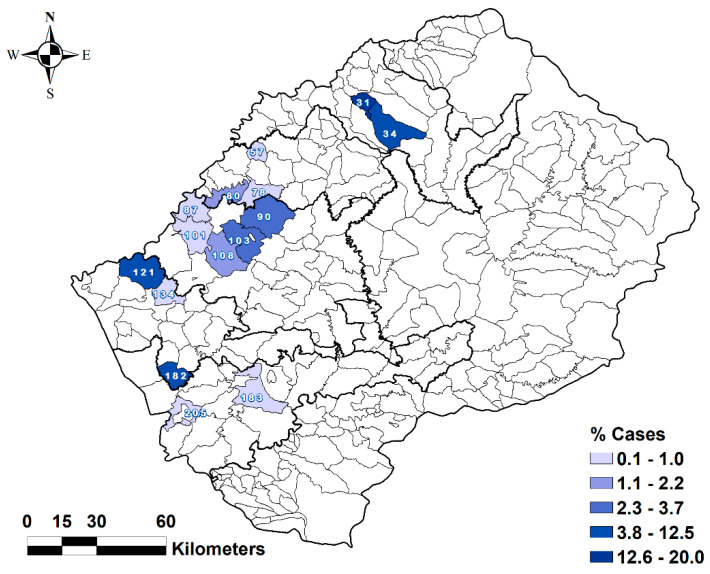
Distribution of anthrax cases over the population at risk in the Lowlands villages of Lesotho from 2005 to 2016.

**Table 1 ijerph-17-07584-t001:** Proportions of anthrax outbreaks in Lesotho by district and village between 2005 and 2016 (N = 38).

District	Village	Anthrax Outbreaks	
		% (Number)	95% CI ^a^
Maseru		53 (20)	37.3–67.5
	Maseru Urban	18 (7)	9.2–33.4
	Mofoka	13 (5)	5.8–27.3
	Ratau	16 (6)	7.4–30.4
	Popa	3 (1)	0.5–13.5
	Mazenod	3 (1)	0.5–13.5
Mafeteng		19 (7)	9.2–33.4
	Kolo	3 (1)	0.5–13.5
	Thaba-Chitja	8 (3)	2.8–20.8
	Boleka	8 (3)	2.8–20.8
Mohale’s Hoek		13(5)	5.8–27.3
	Qalakheng	11 (4)	4.2–24.1
	Qhobong	3 (1)	0.5–13.5
Berea		11 (4)	4.2–24.1
	Khamolane	3 (1)	0.5–13.5
	Thupa-Kubu	5 (2)	1.5–17.3
	Malotoaneng	3 (1)	0.5–13.5
Leribe		6 (2)	1.6–17.3
	Pitseng	3 (1)	0.5–13.5
	Mahobong	3 (1)	0.5–13.5

^a^ 95% Confidence Interval.

**Table 2 ijerph-17-07584-t002:** Proportion of anthrax cases over the population at risk per district and village from 2005 to 2016 in Lesotho.

District	Villages	Anthrax Cases	
		% (*n*/N)	95% CI ^a^
Maseru		1.3 (369/29,070)	1.2–1.4
	Maseru Urban	0.2 (14/8278)	0.1–0.3
	Mofoka	2.0 (211/10,729)	1.7–2.3
	Ratau	2.0 (133/6900)	1.6–2.3
	Popa	3.7 (6/163)	1.7–7.8
	Mazenod	0.2 (5/3000)	0.1–0.4
Berea		0.5 (51/11,208)	0.3–0.6
	Khamolane	2.2 (1/46)	0.4–11.3
	Thupa-Kubu	0.4 (45/10,659)	0.3–0.6
	Malotoaneng	1.0 (5/503)	0.4–2.3
Mafeteng		0.9 (73/8530)	0.7–1.1
	Kolo	10.5 (21/200)	7.0–15.5
	Thaba-Chitja	7.7 (33/430)	5.5–10.6
	Boleka	0.2 (19/7900)	0.2–0.4
Leribe		14.3 (3/21)	5.0–34.6
	Pitseng	12.5 (2/16)	3.5–36.0
	Mahobong	20 (1/5)	3.6–62.4
MohalesHoek		0.2 (30/14,558)	0.1–0.3
	Qalakheng	0.3 (27/10,058)	0.2–0.4
	Qhobong	0.1 (3/4500)	0.0–0.2

^a^ 95% Confidence Interval.

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
