# Peer review of "Spatial Patterns of Anthrax Outbreaks and Cases among Livestock in Lesotho, 2005–2016"

_ijerph, 2020, doi:10.3390/ijerph17207584_

Round 1

Reviewer 1 Report

In "Spatial Patterns of Anthrax Outbreaks and Cases Among Livestock in Lesotho, 2005-2016" the authors map anthrax outbreaks among livestock and identify the Lowlands report all outbreaks.

The paper is interesting and well written.  I have a few suggestions that may improve the paper.

Line 31-33:  Were formal abattoirs inspected and they are no longer inspected?  Please expand on this since it's highlighted as an important change in the paper. 

Comment 1:  Did outbreaks of anthrax change once the formal abattoir closed? You don't mention human cases so it's unclear the link.  I'm assuming all animal cases occur on farm.

Line 32:  Slaughter "slabs".  I'm not sure what is ment here.

Line 57-58:  If the formal abattoir closed, what is an illegal slaughhouse?

Line 79-91:  Was other data available?  Species the anthrax was identified in and date?   This would be very interesting to analyze.

Comment 2:  Contrasting outbreaks to rainfall by year would be interesting.  It may inform better control measures based on environmental changes.

Comment 3:  Are their land management strategies that could reduce animal exposure to anthrax?  The use of drier pastures during peak anthrax exposure times?

Comment 4:  Banning selling products is a very extreme response.  Could better surveillance and testing provide another method of control while not impacting farm sales?  Do you thinking banning sales would just push sales underground?

Comment 5:  If formal abattoirs are not available to identify infected animals, could a public awareness campaign improve disease control while limiting economic impacts?

Line 154:  Change "zooning" to zoning.

Author Response

Comment

Line 31-33:  Were formal abattoirs inspected and they are no longer inspected?  Please expand on this since it's highlighted as an important change in the paper.

Response

The main abattoir has a resident state veterinarian whereas the informal slaughter facilities are not inspected on a day to day bases, it maybe once a month. Line 31-33 has been edited. The main abattoir was not shutdown but temporarily closed between 2003-2008.

Comment

Did outbreaks of anthrax change once the formal abattoir closed? You don't mention human cases so it's unclear the link.  I'm assuming all animal cases occur on farm.

Response

Although, the objectives of this study was not to investigate human cases of anthrax in Lesotho, reference is made to Seeiso and McCrindle (2009) who reported human anthrax cases between 2004 and 2008 as a way to emphasize the public health importance of the disease in the country. Moreover, the time of the cases coincides with the time when the data was collected Regarding the link between the closure of the formal abattoir and outbreaks, using the current data we are unable to investigate this hypothesis.

Cattle keeping in Lesotho is done on a communal basis with almost every household or homestead owning animals. It is not like in the west where you find farms out in the countryside. Here even homes close to the town or city do own animals/cattle, and so cases are not restricted to what you would call a farm.

Comment

Line 32:  Slaughter "slabs".  I'm not sure what is meant here.

Response

Slaughter "slabs" are generally informal slaughter facilities. “Slabs” has been changed to informal slaughter facilities.

Comment

Line 57-58:  If the formal abattoir closed, what is an illegal slaughterhouse?

Response

The temporary closure of the abattoir opened a window for slaughter of animals from unregistered abattoirs also known as informal slaughter facilities. However, some butcheries imported meat from the neighboring South Africa to meet the high demand. We have made modification to the sentence to address the comments of the reviewer.

Comment

Line 79-91: Was other data available?  Species the anthrax was identified in and date?   This would be very interesting to analyze.

Response

We thank the reviewer for comments, in this study we focused mainly on spatial patterns anthrax and cases. There is another paper by the same authors that addresses time and species (Please see reference below).

Lepheana, R.J., Oguttu, J.W. and Qekwana, D.N., 2018. Temporal patterns of anthrax outbreaks among livestock in Lesotho, 2005-2016. PloS one, 13(10), p.e0204758.

Comment

Contrasting outbreaks to rainfall by year would be interesting.  It may inform better control measures based on environmental changes.

Response

We thank the reviewer for comments, in this study we focused mainly on spatial patterns anthrax and cases. This was dictated by the available data. There is however, another paper by the same authors that addresses seasonality in anthrax outbreaks and cases in Lesotho. Please see below.

Lepheana, R.J., Oguttu, J.W. and Qekwana, D.N., 2018. Temporal patterns of anthrax outbreaks among livestock in Lesotho, 2005-2016. PloS one, 13(10), p.e0204758.

Comment

Are their land management strategies that could reduce animal exposure to anthrax?  The use of drier pastures during peak anthrax exposure times?

Response

We thank the reviewer for the suggestion. At the moment we are not aware of any land management strategies that could be implemented reduce animal exposure. We have noted the suggestion by the reviewer and will include this in future studies on potential control strategies for anthrax in Lesotho.

Comment

Banning selling products is a very extreme response.  Could better surveillance and testing provide another method of control while not impacting farm sales?  Do you thinking banning sales would just push sales underground?

Response

The authors agree with the reviewer that banning of products sales is an extreme response. However, surveillance methods require capacity for testing and approval of products which the country does not have. Take for example, wool and mohair which is the biggest industry that gets affected by the ban of sales, Lesotho does not have a wool testing bureau and the alternative is the treatment of products prior to sale during the outbreak and these come at a high cost.

Comment 

If formal abattoirs are not available to identify infected animals, could a public awareness campaign improve disease control while limiting economic impacts?

Response

Thank you for the comment and we have added the following statement in the discussion “In view of this, the Ministry of health  and Ministry of Agriculture and Food Security jointly should development and implement anthrax awareness campaigns to strengthen public awareness in Lesotho.

Comment

Line 154:  Change "zooning" to zoning.

Response

Zooning has been corrected to zoning.

Reviewer 2 Report

This study regarding the spatial analysis of the incidence of anthrax revealed knowledge that is already known that anthrax and other diseases are more common in low-lying areas, but more importantly, this study did not suggest any methods of preventing anthrax or other diseases in livestock.   Simon, et. al. first recorded that abortions in cattle are more common in low-lying areas 1. Later they showed that the cause was associated with nitrate toxicosis.

Recently, Swerczek and Dorton confirmed Simon, et. al. findings that abortions were more common in low-lying areas and the etiology was also associated with nitrate toxicosis. In addition, these workers found many other diseases of livestock were also more common in low-lying areas and absent in higher areas on the same farm. Similar findings were found with your spatial anthrax findings.

The reason why diseases of livestock, including anthrax, is more common in low-lying areas is because soil-types are rich in nutrients, including nitrates, sulfur, and potassium. These nutrients in excess will exacerbate most all diseases of livestock. These and other nutrients come from higher elevations to settle in low-lying areas. In addition, any water accumulation in low-lying areas and in water holes are also high in nitrate, potassium, and sulfur and other nutrients that exacerbate infectious diseases of livestock and wildlife. In these areas, water in water holes become more toxic during droughts as the potentially toxic nutrients become more concentrated.

It is obvious that livestock grazing in low-lying areas, other supplementations of nutrients are contraindicated. In addition, it is imperative that adequate salt is available at all times. Seemingly, sodium neutralizes excessive nitrate and potassium. This is discussed in the Swerczek and Dorton publication 2.

Possibly the authors want to suggest reasons why anthrax is more common in low-lying areas and methods of control, otherwise, the paper is of little value other than anthrax is more common in low-lying areas, that is already known

  1. Simon, J. et. al. Pathological changes associated with the lowland abortion

        syndrome in Wisconsin. JAVMA. 1958, 132, 164-9

  1. Swerczek, T.W. and Dorton, A.R.   Effects of Nitrate and Pathogenic Nanoparticles on Reproductive Losses, Congenital Hypothyroidism and Musculoskeletal Abnormalities in Mares and Other Livestock: New Hypotheses. Animal and Veterinary Sciences. 2019, 7, 1-11.   doi: 10.11648/j.avs.20190701.11

Author Response

Comments and Suggestions for Authors

This study regarding the spatial analysis of the incidence of anthrax revealed knowledge that is already known that anthrax and other diseases are more common in low-lying areas, but more importantly, this study did not suggest any methods of preventing anthrax or other diseases in livestock.  

Response

We thank the reviewer for the comment. However, it is important to note that this is the first study to investigate spatial patterns of anthrax in Lesotho. Furthermore, this is the first to show that due to its spatial disparity it is possible to control anthrax outbreaks in Lesotho using the zoning system which enables trading in animals and animal products even in the presence of an outbreak. This in our view is novel -it has never been observed that in countries like Lesotho that can distinctly be divided into Highlands and Lowlands that zoning could be contemplated as a control strategy. Furthermore, what the reviewer is referring to by saying that we know that the disease is more common in low-lying areas is not the main finding in this study. The Lowlands in this study are not synonymous to low laying areas. Within the Lowlands one could have low laying areas. And as we explain, we were not able to confirm the theory that in Lesotho’s Lowlands the disease occurs more predominantly in the low laying areas. 

Comment

Simon, et. al. first recorded that abortions in cattle are more common in low-lying areas 1. Later they showed that the cause was associated with nitrate toxicosis. Recently, Swerczek and Dorton confirmed Simon, et. al. findings that abortions were more common in low-lying areas and the etiology was also associated with nitrate toxicosis. In addition, these workers found many other diseases of livestock were also more common in low-lying areas and absent in higher areas on the same farm. Similar findings were found with your spatial anthrax findings. The reason why diseases of livestock, including anthrax, is more common in low-lying areas is because soil-types are rich in nutrients, including nitrates, sulfur, and potassium. These nutrients in excess will exacerbate most all diseases of livestock. These and other nutrients come from higher elevations to settle in low-lying areas. In addition, any water accumulation in low-lying areas and in water holes are also high in nitrate, potassium, and sulfur and other nutrients that exacerbate infectious diseases of livestock and wildlife. In these areas, water in water holes become more toxic during droughts as the potentially toxic nutrients become more concentrated. It is obvious that livestock grazing in low-lying areas, other supplementations of nutrients are contraindicated. In addition, it is imperative that adequate salt is available at all times. Seemingly, sodium neutralizes excessive nitrate and potassium. This is discussed in the Swerczek and Dorton publication 2. Possibly the authors want to suggest reasons why anthrax is more common in low-lying areas and methods of control, otherwise, the paper is of little value other than anthrax is more common in low-lying areas, that is already known.

Simon, J. et. al. Pathological changes associated with the lowland abortion         syndrome in Wisconsin. JAVMA. 1958, 132, 164-9

Swerczek, T.W. and Dorton, A.R.   Effects of Nitrate and Pathogenic Nanoparticles on Reproductive Losses, Congenital Hypothyroidism and Musculoskeletal Abnormalities in Mares and Other Livestock: New Hypotheses. Animal and Veterinary Sciences. 2019, 7, 1-11.   doi: 10.11648/j.avs.20190701.11

Response

The authors thank the reviewer for the comments and agree that it is possible that soil type could be playing a role in anthrax outbreaks in this study. This phenomenon has been discussed in much more details in a number of studies. However, we think that the reviewer interpreted Lowlands in our study to mean low laying areas. The two are not synonymous. We would like to remind the reviewer that in the methodology we did indicate that for purposes of this study the country was divided into two: the Lowlands and the Highlands. We then observed that the disease exclusively occurs in the Lowlands and went ahead and suggested implementation of zoning as suitable control strategy to mitigate the impact of the disease especially on trade. We also acknowledge that it is possible that in Lowlands, the disease does also occur more predominantly but not exclusively in the low laying areas of the Lowlands. However, we could not confirm this because we did not have the data to do so.

Furthermore, soil type is not the only possible determinant of anthrax outbreaks in the Lowland areas. Temperature, humidity, presence of catchment which facilitates the accumulation of B. anthracis spores are also important factors to consider. However, since data on these potential predictors was not available in this study this could not be assessed.

Round 2

Reviewer 1 Report

The paper is acceptable in the revised form.